# Insight into the Lubrication and Adhesion Properties of Hyaluronan for Ocular Drug Delivery

**DOI:** 10.3390/biom11101431

**Published:** 2021-09-30

**Authors:** Mikuláš Černohlávek, Martina Brandejsová, Petr Štěpán, Hana Vagnerová, Martina Hermannová, Kateřina Kopecká, Jaromír Kulhánek, David Nečas, Martin Vrbka, Vladimir Velebný, Gloria Huerta-Angeles

**Affiliations:** 1Department of Applied Chemistry (R&D), Contipro a.s., Dolní Dobrouč 401, 561 02 Dolní Dobrouč, Czech Republic; mikulas.cernohlavek@vutbr.cz (M.Č.); Martina.Brandejsova@contipro.com (M.B.); Petr.Stepan@contipro.com (P.Š.); Hana.Vagnerova@contipro.com (H.V.); Martina.Hermannova@contipro.com (M.H.); Katerina.Kopecka@contipro.com (K.K.); Jaromir.Kulhanek@contipro.com (J.K.); Vladimir.Velebny@contipro.com (V.V.); 2Department of Tribology, Faculty of Mechanical Engineering, Brno University of Technology, Technická 2896/2, 616 69 Brno, Czech Republic; necas@fme.vutbr.cz (D.N.); Martin.Vrbka@vut.cz (M.V.)

**Keywords:** hyaluronan, biotribology, friction, lubrication, eye drops

## Abstract

Hyaluronan (HA) is widely used for eye drops as lubricant to counteract dry eye disease. High and low molecular weight HA are currently used in ophthalmology. However, a large portion of the current literature on friction and lubrication addresses articular (joint) cartilage. Therefore, eye drops compositions based on HA and its derivatized forms are extensively characterized providing data on the tribological and mucoadhesive properties. The physiochemical properties are investigated in buffers used commonly in eye drops formulations. The tribological investigation reveals that amphiphilic HA-C12 decreases the friction coefficient. At the same time, the combination of trehalose/HA or HAC12 enhances up to eighty-fold the mucoadhesiveness. Thus, it is predicted a prolonged residence time on the surface of the eye. The incorporation of trehalose enhances the protection of human keratinocytes (HaCaT) cells, as demonstrated in an in-vitro cell-desiccation model. The presence of trehalose increases the friction coefficient. Medium molecular weight HA shows significantly lower friction coefficient than high molecular weight HA. This research represents a first, wide array of features of diverse HA forms for eye drops contributing to increase the knowledge of these preparations. The results here presented also provide valuable information for the design of highly performing HA-formulations addressing specific needs before preclinic.

## 1. Introduction

There are over 140 million contact lens wearers worldwide, and many of these wearers suffer from signs and symptoms of dry eye disease (DED), including discomfort, dryness, and red eyes [1]. Clinical studies have shown a prevalence of discomfort ranging from 10 to 50% and uncomfortable hours of wear in contact lens wearers. Of the symptoms reported, sensation of dry eye is the most common [2]. Furthermore, contact lens wearers complain of various types of contact lens discomfort. Eye drops are the first-line treatment for many causes of ocular irritation by reducing friction between the eyelids and the cornea [3]. Ophthalmic demulcents (water soluble polymers) are vital components of both over-the-counter (OTC) and prescription formulations used for managing dry eye and for caring for contact lenses. They lubricate the epithelium and decrease surface irritation. Between the used demulcents, Hyaluronic acid (HA, hyaluronan) is a high-molecular-weight glycosaminoglycan composed of D-glucuronic acid (GlcA) and *N*-acetyl-D-glucosamine (GlcNAc) units linked by alternating β-(1 → 4)-and β (1 → 3)-glycosidic bonds. HA is found in various tissues in the eye, is biodegradable and biocompatible, and it is one of the most important and well-studied biomolecules of the extracellular matrix. HA is one of the most used ingredients in artificial tears and multipurpose solutions as wetting or comfort agent. HA can be classified into five groups according to its molecular weight: Very high molecular weight (vHMM-HA > 5000 kDa), high molecular weight HA (HMW-HA, 3000–1000 kDa), medium molecular weight HA (MMW-HA, 1000–250 kDa), low molecular weight HA (LMW- HA, 250–10 kDa) and oligosaccharides (<10 kDa) [4]. The molecular size dictates its physiochemical properties, but it also dictates its biological activity [5,6]. In fact, the viscoelastic properties of any preparation (composition) containing native and/or modified HA will be highly influenced by HA concentration, Mw and chemical or physical modification. In the case of eye drops, the Mw of HA is usually not reported by the manufacturers [7]. Some studies had used vHMM-HA [5]. However, eye drops are usually autoclaved prior to use, therefore, the average molecular weight of HA decreases by hydrolysis. The ideal HA-based eye drops should probably include HMW-HA, which warranty an enhanced viscosity at low shear rate without exceeding the blur threshold [7]. 

The coefficient of friction (CoF) at a variety of biointerfaces can be determined using tribology. For example, Morrison et al. measured the in-vitro boundary lubrication effects of Proteoglycan 4 (PRG4 or lubricin) and HA. The authors characterized the in-vitro boundary lubrication properties at resected cadaveric human corneas—polydimethylsiloxane (PDMS) interface. Lubricin was suspended either in saline solutions or in AQuify Comfort eye drops (HA 0.1%). Thus, PRG4 (300 µg/mL) and the 0.1% HA containing eye drops functioned synergistically to further reduce friction [8]. Samson and collaborators tested the boundary lubricating ability of Blink contacts^®^ eye drops (HA, 0.15%) combined with lubricin on contact lenses. Incorporation of both HA and PRG4 to form an HA/PRG4 complex produced synergistic properties that reduces friction [9].

Sterner et al. compared the frictional properties of HA and PVP (polyvinylpyrrolidone) as a function of structural parameters such as Mw, concentration, and affinity to the substrate. The polymer chain length can impact the frictional properties of polymer brushes. Increasing the molecular weight of poly(N-vinylpyrrolidone) (PVP) brushes from 10 to 1300 kDa led to two orders of magnitude decrease in CoF between polydimethylsiloxane (PDMS) and silicon wafer interphases. Besides, HA’s enhanced boundary lubricating abilities was noted when it was chemisorbed on ocular tissue [10]. Yamasaki and collaborators compared the adsorption of hydrophobized-HA derivative (HAD) and LMW-HA to four soft contact lens materials. HAD deposited more readily to silicon hydrogel contact lenses than lenses incubated with native HA. These results can likely be attributed to the difference of the chemical structure of the biopolymer (amphiphilicity) [11]. 

Previous studies of the relationship between the rheological properties of HA solutions and their clinical efficacy have not been definitive. Furthermore, it is difficult to identify the effect of the components as the composition of commercially available eye drops is complex. 

On the other hand, the lubrication function of HA solutions is determined by their structure, e.g., length of the chain, macromolecular architecture, composition, and adhesion to the ocular surface. The aim of this study was to evaluate the importance of molecular weight and lubricant composition (phosphate, HEPES buffer vs complex trehalose buffer) on the lubricating performance of native HA versus hydrophobized (sodium dodecanoyl -hyaluronan). Such information may aid in the development or improvement of biomolecules used as soft lubricated interfaces, with focus on the eye. The efficiency of HA as a boundary lubricant has been disputed in the literature [12]. HA appears to have a stronger wear-reducing role than a friction-reducing one under conditions like those in articular joints [13]. However, the tribological models included low friction at the high pressures of the joints. While typical value of the eyelid pressure on the cornea at rest being around 1 kPa. Regarding the eyelid’s sliding speed, which depends on a diverse set of factors such as age and gender, eye health status and blinking, it is estimated that the speed can reach values as high as 100–200 mm/s [14].

A tribological investigation is performed to evaluate the lubrication properties of HA by using contact lenses as in-vitro model. The coefficient of friction (CoF) is evaluated as a function of buffers composition. Furthermore, the in-vitro efficacy of OTC products, HA and HA-C12 for their ability to protect human epithelial cells (HaCaT) from dehydration is studied.

On the other hand, key components of the protective tear film are mucins, large glycoproteins that serve as molecular lubricants on many epithelial body surfaces [15]. Therefore, the interaction mucin/HA is studied in the presence of ingredients usually present in OTC products. Such information may aid in the development or improvement of biomolecules used as soft lubricated interfaces with focus on ocular drug delivery. A detailed molecular-level understanding of the effect of the macromolecular structure have benefits ranging from better treatments of friction-related joint diseases to improved devices including contact lenses, where low friction is required.

## 2. Materials and Methods

### 2.1. Materials

Hyaluronan of several molecular weights (Mw) and polydispersities (PDI = Mw/Mn) in brackets) was used: HA0 = 260 (1.5), HA1 = 300 (1.5), HA2 = 440 (1.6), HA3 = 589 (1.5), HA4 = 848 (1.6), HA5 = 953 (1.6), and HA6 = 1656 (1.6) kDa, respectively. HA was obtained by fermentation and was provided by Contipro a. s (Dolni Dobrouč, Czech Republic). Mucin type II and III (bound sialic acid 0.5–1.5%, partially purified powder), 4-(2-hydroxyethyl) piperazine-1-ethanesulfonic acid (HEPES, 99.5%), were obtained from Sigma-Aldrich (Czech Republic). D-(+)-trehalose dihydrate (≥98%) was obtained from TCI-Europe (Zwijndrecht, Belgium). Sodium chloride (NaCl, 98%), tetrahydrofuran (99.0% THF), sodium hydroxide (NaOH, 98%), and hydrochloric acid (HCl, 35%) were obtained from Lachner Ltd. (Neratovice, Czech Republic). HEPES buffer (pH 7.0) was prepared by dissolving HEPES (4.77 g) in 1 L of distilled water. The pH 7.2 was adjusted with 1 M NaOH. The trehalose buffer contains trehalose (3%), tris (hydroxymethyl) aminomethane 0.12%, NaCl (0.2%). The pH was adjusted to pH 7.2. by HCl. The physiological buffer contains NaCl (0.9%).

Ten over-the-counter eye drops were evaluated: Systane Hydration^®^ (HA 0.15%, polyethylene glycol 400 kDa 0.4%, propylenglycol 0.3%, hydroxypropyl guar, Alcon, Fort Worth, TX, USA), Laim Care^®^ (HA 0.30%, Schalcon, Rome, Italy), Thealoz Duo^®^ (HA 0.15%, trehalose 3%, NaCl, trometamol, Théa Laboratories, Clermont-Ferrand, France), Refresh^®^ (Carboxymethylcellulose 0.5%, Allergan, Irvine, CA, USA), ReNu^®^ (PVP, boric/borate, KCl, NaCl, edetate disodium 0.1%, sorbic acid, Bausch & Lomb Incorporated, Waterford, Ireland), Ocutein sensitive^®^ (0.10 %, DaVinci Academia, Simply You Pharmaceuticals a.s., Prague, Czech Republic), Hyabak^®^ (0.15%, Laboratorios Thea, Clermont-Ferrand, France), Vismed^®^ (0.18%, TRB Chemedica, Newcastle-under-Lyme, UK), Hylo-gel^®^ (0.2%, Ursapharm, Saarbrucken, Germany) and hyal-drop multi^®^ (HA 0.24%, Bausch & Lomb, Prague, Czech Republic). All the measurements were performed up to 24 h after opening the package. Bioinfinity^®^ (Cooper Vision Inc., San Ramon, CA, USA), silicone hydrogel soft contact lenses were used for the tribological measurements. The lenses contain 48% water and 52% Comfilcon A.

### 2.2. Synthesis of Sodium Dodecanoyl Hyaluronate (HA-C12)

The derivatives tested in this work were prepared by acylation of sodium hyaluronate in a mixed water/THF solvent. The chemical modification of HA grafted by dodecanoyl fatty acid moieties was reported elsewhere [16,17]. 

### 2.3. Structural Characterization

The degree of substitution (*DS_GC_*), i.e., the content of esters moieties (acyls), was measured by gas chromatography after alkaline hydrolysis of the sample as described recently [17]. The *DS_GC_* means the extent of HA modification by hydrophobic side groups and is calculated according to the Equation (1): (1)$$DS_{GC}=\frac{m_{FA,tot}-m_{FA,free}}{m_{sample}}\frac{M_{HA}}{M_{FA}}\times 100$$
where *m_FA,tot_* is the amount of fatty acid determined in the derivative after alkaline hydrolysis, *m_FA,free_* is the amount of fatty acid unbound to *HA* chain, *m_sample_* is the weight of derivative, *M_HA_* is molar mass of *HA* disaccharide (401 g/mol for sodium salt), and *M*_FA_ is molar mass of fatty acid. 

### 2.4. Size Exclusion Chromatography–Multiangle Laser Light Scattering (SEC-MALLS) 

The molecular weight of HA and HA-C12 was determined by SEC-MALLS on a Waters Alliance liquid chromatography (Model e2695), equipped with a refractive index detector (Model 2414, Waters, Santa Clara, CA, USA). The chromatograph is provided by a mini DAWN TREOS multiangle laser light scattering photometer (Wyatt Technology Corporation, Dernbach, Germany). The injection volume was 100 µL of HA (0.02%). The separation was carried out using two columns thermostated at 40 °C: PL aquagel-OH 60 (7.5 × 300 mm, 8 µm) and PL aquagel-OH mixed-H (7.5 × 300 mm, 8 µm). Each sample was filtered across an acrodisc syringe filter (0.22 µm, 13 mm diameter) with the support membrane (Pall corporation, New York, NY, USA). An aqueous solution of NaH_2_PO_4_ (50 mM, pH adjusted to 7.5) was filtered through a Nylaflo^®^ (nylon) membrane filter (0.2 µm, 47 mm diameter, Pall corporation, New York, NY, USA) was used as a mobile phase. Data acquisition and molecular weight calculations were performed using the ASTRA V software (Wyatt Technology Corporation). A specific refractive index increment dn/dc = 0.155 mL/g was used for HA [18,19].

### 2.5. Nuclear Magnetic Resonance Spectroscopy 

^1^H spectrum was recorded at 25 °C using a 700 MHz BRUKER Avance TM III. High molecular weight HA samples were analyzed by dissolving the derivatives (8 mg/mL) in D_2_O. 

### 2.6. Tribological Setup

A commercial tribometer Bruker UMT TriboLab (Billerica, MA, USA) was used in a pin-on-plate configuration undergoing reciprocating linear motion. To evaluate the friction properties in an eye-mimicking model a pin (eyeball model) was used, considered a common model for tribological measurements [10]. The silicone rubber ball (Elastosil^®^ LR 3003, hardness = 10 Sh, E = 300 kPa) with diameter of 17.2 mm (respecting radius of the cornea of 8.6 mm) with the contact lenses on its surface. Vulcanized silicone rubber Elastosil^®^ LR 3003 was chosen with respect to the known natural corneal material properties (Young’s modulus E), which are in accordance with the study of Shih et al. [20]. Bioinfinity^®^ contact lenses were used for the tests. The plate was used as the eyelid model and simulated using a silicone rubber (Elastosil^®^ LR 3003, hardness = 30 Sh, E = 760 kDa) with dimensions 100 × 65 × 35.5 mm. The contact lens was moved to the pin to mimic the lens’s conditions embedded in the eye before the testing. A 50 N sensor DFH-5-G having a resolution of 2.5 μN was used. The contact was lubricated with commercial eye drops, HA or HA-C12 solutions at various concentrations (0.1–0.3%). The experiments consisted of 60 s running-in phase and two subsequent friction tests lasting 300 s. Coefficient of friction (CoF), was evaluated as a function of time from 10 independent values. CoF defined as the ratio between friction force and normal force. The final values were obtained by sample averaging. Both the contact lenses and the measured solution were replaced with new samples after each test. The pin was loaded with a normal force of 1 N and fixed in a stainless-steel holder. The holder performed a reciprocating motion with a frequency of 6 Hz. The average sliding speed for experiments was calculated to be 156 mm/s, corresponding to physiological blinking [21,22]. The stroke of the experiment was 13 mm. The contact was maintained at 37 °C by heating cartridges (2 × 145 W, Hotset, Pilsen, Czech Republic). The temperature was regulated using a thermocouple (Hotset). CoF represents the average of two independent determinations measured five times for each sample.

### 2.7. Mucoadhesive Index

The mucoadhesive index of the HA/HAC12 solutions was evaluated by rheology.

The mucoadhesive index was defined as: ∆ (%) = [ηmix − (ηmuc+ηHA − C12)]/((ηmuc + ηHA − C12)) × 100(2)
where Δ (%) is the mucoadhesion index, ηmuc is the dynamic viscosity of mucin, ηHAC12 is the dynamic viscosity of the sample, and η mix is the apparent viscosity (mPa·s) of the mixture HA + mucin.

### 2.8. Evaluation of Protective Efficacies of Different Lubricants on HaCaT Cells against Dehydration

Commercially available eye drops, HA and HA-C12 were tested for their ability to prevent the death of HaCaT cells (human epidermal keratinocyte lines) exposed to dehydrating conditions. HaCaT cells are a valuable alternative to primary epithelial cell lines to assess viability of formulations for ocular use [23]. The HA and HA-C12 samples were sterilized by filtration. The solutions were filtered using an Ophtalsart^®^ filter (0.2 µm, Göttingen, Germany). Experiments were performed using 0.3% solutions of the lubricants (HA, HA-C12) diluted in trehalose or HEPES-buffers for the protection against dehydration assays. OTC products were tested as well under similar conditions. The cells were cultured at 37 °C in an atmosphere of 5% CO_2_ in Dulbecco’s Modified Eagle’s Medium (DMEM). After reaching 100% confluency, the cultivation medium was discarded, thereafter, 150 µL of 0.3% of HA, HA-C12, or OTC sterile solutions was added to the cells. The cells were incubated for 60 min at 37 °C. After removal of the test solutions, the cells were dried in room air for 30 min. The cells were incubated for 2.5 h at 37 °C with DMEM culture medium containing tetrazolium salt (3-(4,5-dimethylthiazol-2-yl)-2,5-diphenyltetrazolium bromide (MTT) reagent. Then all MTT solution was removed and 220 μL of lysis solution (550 µL IPA: DMSO (1:1)) was added. The cells were lysed for 30 min at room temperature on a shaker. The MTT dye is reduced by viable cells to a colored formazan salt as an indicator of the metabolic activity of cells. The absorbance was measured with a microplate reader (VERSAmax, Molecular devices, San Jose, CA, USA) at 570 nm according to the manufacturer’s instructions. For wells that were not exposed to dehydration (positive control), all the cells (100%) survived for the 60-min test. The average survival rates for cells exposed to dry conditions (negative control) was determined to be around 10%. The survival rate of the cultured cells was calculated according to the following formula: Cell survival rate = (Absorbance of lubricant pretreated/Absorbance of dry control) × 100(3)

The results are reported as the mean (average value). Data were analyzed using one-way ANOVA or the *t*-test as appropriate. Results with *p* < 0.05 seemed as significant. All experiments were carried out at least in six independent replicates.

## 3. Results and Discussion

### 3.1. Synthesis and Characterization of Amphiphilic Hyaluronan (HA-C12)

The amphiphilic HA-C12 is prepared using medium molecular weight HA by mixed anhydrides chemistry [17]. Dodecanoic acid is activated by benzoyl chloride (BC) in tetrahydrofuran (Figure 1A). The activation reaction is performed for 30 min. The mixed anhydride reacts with HA yielding the esterified product (Figure 1B). The molecular weight of HA in the reaction feed is considered an important parameter affecting the esterification reaction. Low molecular weight HA (Mw~260 kDa) is more reactive than medium molecular weight HA (Mw~589 kDa). The chemical modification of HA with Mw higher than 600 kDa reaches low degree of substitution (DS ≤ 5% mol/mol). Thus, these results cannot be used as comparison in this work. Figure 2 depicts the ^1^H NMR spectrum of HA-C12 recorded in deuterated water and NaOD. The addition of NaOD improves the spectral resolution. While the ^1^H NMR spectrum of HA-C12 recorded in D_2_O derivatives presented a broadening of the signals due to self-aggregation. The degrees of substitution determined by NMR in NaOD agree with the ones determined by gas chromatography (Table 1, DS_GC_). However, the signals of possible free (unbound) dodecanoic acid can be overlapped with a possible overestimation of the degree of substitution. Figure 2A shows the ^1^H NMR spectrum of HA-C12 in D_2_O. The spectrum shows typical proton chemical shifts of HA with signals at 2.0 ppm (-NCOCH_3_ group), backbone signals at 3.4–3.9 and anomeric resonances from 4.4 to 4.6 ppm. The signal corresponding to C2 is shielded from 2.4 ppm to 2.2 ppm. Moreover, the signal at 1.6 ppm (C3) shifts to 1.5 ppm. Figure 2B shows peaks at 0.8, 1.3, 1.6 and 2.4 ppm, which are attributed to CH_3_; CH_2_ in the fatty acid chain.

Amphiphilic derivatives interact strongly with the column, impairing the determination of Mw by SEC-MALS [24]. Thus, HA-C12 is analyzed using a mobile phase H_2_O/isopropyl alcohol 60/40 (*v*/*v*) using NaH_2_PO_4_ (50 mM) as buffer at 40 °C. A decrease of molecular weight is observed after chemical modification. For example, when low molecular weight HA is modified (Mw~260 kDa), thus, the esterified product presents a lower Mw (Mw~200 kDa) and low polydispersity index.

As the solvent mixture drastically influences the selectivity of the reaction towards selected substitution at the primary and/or secondary hydroxyl moieties in HA [16]. The even substitution drastically changes the conjugate’s solubility. Therefore, the derivatives are prepared at low concentration (1.0% of HA in the reaction feed) to obtain fully water solubility. The solubility is expected for solution characterized by low turbidity and higher transmittance, T ≥ 65%). The solubility of the derivative is explained in terms of regioselectivity of the dodecanoyl substitutions at C6 and C4, which influenced distinctively the backbone flexibility in aqueous media. Therefore, unsaturated C12-modified tetrasaccharides (ΔHA4-1 × C12 and ΔHA4-2 × C12) are the two main SpHyl degradation products of HA-C12. The extracted ion chromatograms of the fragments are shown in Appendix A. Particularly, the derivative is characterized by 33% of disubstituted tetramers (ΔHA4-2 × C12) and 50% of monosubstituted ones (ΔHA4-1 × C12). The experimental results showed that a higher proportion of di-substitution (50–100%) would produce insoluble derivatives. As the degradation of HA requires long time, transmittance and turbidity of the prepared solutions is used as criteria of solubility. For eye drop solutions, the expected transmittance should be higher than 92% [25]. Additionally, the turbidity of the solutions is lower than 10 NTU, which confirms that the samples do not contain suspended particles. 

### 3.2. Tribological Model of the Eye

The eye is a complex organ with several compartments separated by various tissues, including highly cellular layers and acellular liquid or gel-like matrix, thus it is difficult to be studied in-vitro. Even though, several works used cadaveric eyes, the physiology and parameters of the eye change due to vitreous liquefaction, resulting in data that differed by orders of magnitude [26]. Therefore, the designed tribological model of the eye consists of a silicone ball, a contact lens, and a silicone plate (pin-on-plate configuration) as depicted in the Figure 3A. The material combination (Section 2.6) is defined after preliminary experiments. The optimal average coefficient of friction (Figure 3B) is obtained with an applied normal force of 1 N. The 10 Sh ball exhibits the lowest standard deviation from all measured samples indicating a good indication of the stability and repeatability of the setup. The Bioinfinity^®^ contact lenses show the highest experimental reproducibility. The used pin-on-plate configuration well mimicked the blinking mechanism with a reciprocating motion of 6 Hz. The eyelid’s sliding speed at blinking reported between 100 to 200 mm/s [14]. In this work, the average sliding speed is 156 mm/s.

### 3.3. Determination of the Coefficient of Friction for Commercially Available Eye Drops

The frictional properties of HA and demulcents are evaluated on Bioinfinity^®^ contact lenses (Figure 4). This experiment elucidates the ability of the demulcents to reduce boundary friction as a function of chemical structure and affinity to the substrate. Furthermore, hydrophilic lenses are a relevant model [27], as in-vitro contact lenses’ friction significantly correlates with contact lens discomfort [28]. Systane-Ultra^®^ has a significant lower coefficient of friction (0.054 ± 0.0048) compared to the other lubricants tested. The low CoF is probably explained by the synergy of two demulcents (polyethylene glycol, propylene glycol) and a biopolymer (hydroxypropyl guar), crosslinked by borate ions [29]. Laim-care^®^ (HA 0.3%) has a low CoF (0.058 ± 0.003) and is explained by either the high concentration of HA or the Mw. 

Thealoz duo^®^ (Mw~220 kDa) shows a high CoF (0.077 ± 0.001). The last-mentioned formulation is usually recommended for patients with moderate to severe dry eye syndrome [30,31]. Refresh^®^, a low viscosity formulation, has a high CoF (0.069 ± 0.008). This agrees with a previously reported work, Refresh contacts drops proved to have less effect on the frictional properties of the lotrafilcon B lenses, showing a smaller decrease in friction than Systane^®^ [32]. ReNu^®^ (0.061 ± 0.005) showed a high CoF, which could be related to blurring effect in patients [33].

### 3.4. Determination of the Coefficient of Friction for Native HA

The influence of Mw is tested at the same concentration used in Laim-Care^®^ (Figure 5). The determination of CoF over time shows that samples with medium molecular weight (HA2—440 kDa, HA3—589 kDa and the derivative HA-C12) have more consistent values of CoF over time than the sample with high molecular weight (HA6—1600 kDa). The high molecular weight HA6 has the tendency to decrease over time, but initial and final values of CoF is higher than for any other sample (Figure 5A). The average value of the coefficient of friction for medium molecular weight HA (HA2 and HA3) was around 0.05 (±0.0004). For high molecular weight-HA (HA6), CoF is almost two-fold (0.089 ± 0.006). Similar results are observed for the buffer containing trehalose, where individual samples of HA are compared (Figure 5B). Interestingly, the CoF of HA3 in trehalose-buffer is 0.056 (±0.0007), while HA6 reaches 0.10 (±0.004). Therefore, medium molecular weight HA acts as an excellent lubricant, with low CoF values. Our results are determined at high sliding velocity (156 mm/s) or physiological blinking. In contrast to previous studies, high molecular weight HA-dissolved in phosphate buffer presented a low CoF at low sliding speeds 0.1 mm/s and low loads (0.25 to 4 mN) [10]. But it should be considered that when the eyelid traverses the cornea during physiological blinking, the tribological characteristics transitioned from boundary to hydrodynamic lubrication [21]. 

### 3.5. MucoadhesionProperties of HA Determined by Turbidimetric Titration

The interaction of HA and mucin is evaluated by turbidimetric titration [34,35]. The results are reported in Appendix A. The effect of molecular weight is studied between 300 and 1600 kDa varying concentration. The interaction HA/mucin III is found to be independent of the molecular weight in HEPES buffer. Even though, Salzillo et al. showed HMW-HA (Mw~1100 kDa) is the most mucoadhesive [36]. Still, the high concentration of HA (0.67%) limits its applicability due to the high cost of the formulation. In the case of HA-C12, the interaction was concentration and Mw dependent.

### 3.6. Mucoadhesive Properties Determined by Rheology

Systane Ultra^®^ and Laim Care^®^ exhibit a moderate shear thinning behaviour (Figure 6A). Besides, Thealoz duo^®^, Refresh^®^ and ReNu^®^ present low viscosity at low and high shear rates. From the tested OTC eye drops, the highest mucoadhesiveness is observed for Systane^®^ (Figure 6B).

Rheological characterization of HA2-C12 in HEPES, trehalose (T) and saline buffers is determined at 25° (Appendix A). The derivative exhibits shear thinning behaviour with high viscosity (30.4 mPa·s) at low shear rate and low viscosity (12.0 mPa·s) at high shear rate in T buffer. The viscosity of amphiphilic HA is changing with the media. At the same concentration, a high viscosity is observed (603.9 mPa·s) in saline buffer.

The mucoadhesive index is calculated at shear rate of 33.9 s^−1^ corresponding to physiological blinking [36]. The mucoadhesive index for native HA increases in trehalose buffer (Table 2, entries 3, 4 and 6, 7). The mucoadhesiveness boost with the concentration of HA-C12 (Table 2, entries 8–10 and 12–14). HA-C12 is characterized by eighty-fold higher rheological synergism compared with HA. Amphiphilic HA can be formulated without trehalose with a higher mucoadhesive effect. Mucoadhesion is related to the ocular residence time of the eye drops [37]. The robustness of the method is evaluated by using mucin from porcine stomach, Type II and III in HEPES (Appendix A). Unfortunately, the degradation of HA is found for HA6 with mucin III in HEPES buffer. This fact could be explained by bacterial contamination in commercial mucins. For more accurate results, it is necessary to purify the native mucins before measuring [38]. 

### 3.7. Sterilisation by Filtration

Sterilization remains one of the main issues and needs to be solved before preclinic. As amphiphilic HA is physically crosslinked by interaction of the alkyl chains, thus, the filtration is challenging. In this work, filtration is used for sterilization before in-vitro cellular testing (Section 3.8). HA-C12 solutions are sterilized by filtration using Ophtalsart, a cellulose based-material for sterilization (Appendix A). The high viscosity of HA3-C12 limits its filtration.

### 3.8. Evaluation of Protective Efficacies of HA and Derivatives Tested on HaCaT Cells against Dehydration

HA retains a significant fraction of water within its structure. This property is also thought to contribute to the maintenance of tear film stability and its ability to keep the ocular surface moistened. The synergism of HA/trehalose eye drops was tested in patients after cataract surgery (Thealoz gel^®^). The results were compared to with patients treated with HA alone (Hyabak^®^). Both preparations are formulated with LMW-HA (220 and 248 kDa, respectively) [39]. Even though Caretti et al. showed that eyedrops containing both trehalose, HA and carbomer appeared more effective than HA alone after cataract surgery. The authors pointed that additional studies regarding efficacy of this new formulation have not been published. Thus, the protective effectiveness of HA and against dehydration of cells is evaluated in this work. The synergic effect of trehalose/HA is Mw dependent (Figure 7A). HaCaT cells are protected by HMW-HA (Mw~1600 kDa). The second set of our experiments is designated to evaluate the protective capability of the buffer, HA and HA-C12 as lubricants against dehydration of HaCaT. Again, our results demonstrate that the HaCaT survival rate is significantly higher for cells pretreated with HA-C12 than for those pretreated with HA or OTC products-containing HA, except for Hyaldrop^®^ which contains medium molecular weight 0.24% HA (Mw~533 kDa). Thus, the effectiveness of HA appears as dose dependent [40]. As demonstrated in Section 3.7, HA-C12 interacts with mucin III. Therefore, HA2-C12 might increase the residence time of the formulation in the eye and hence improve the adhesion to the eye surface. Furthermore, amphiphilic HA- might act as a reservoir of hydrophobic molecules [41]. Figure 7B,D show that HA2-C12 protects almost 100% of HaCaT cells against dessication (alike Hyaldrop^®^). The protective effect is higher compared to HA used for chemical modification, which protected only 50% of the cells.

The survival rates of cells pre-treated with OTC products are shown in Figure 7C. Our results demonstrate that HaCaT survival rate is significantly higher for cells pre-treated with Laim-care^®^ or Systane Ultra^®^ (*p* < 0.0001, *n* = 6). These results agree with the enhanced viscosity of the formulations [42]. The Mw of HA formulated in Laim-Care^®^ is determined to be 945.6 kDa with a polydispersity of 1.8 (Appendix A). The data supported that high molecular weight HA is the most effective. These solutions would result in the optimal formulation for providing the longest duration of relief and best comfort for the patient [7]. Thealoz duo^®^ and Hyabak^®^ are not significantly different from their positive controls, indicating their ineffectiveness in protecting cultured HaCaT cells against dehydration. Interestingly, Hyabak^®^ and Thealoz duo^®^ are formulated in different packages, but they are characterized by the same composition and concentration of HA (0.15%) and Mw. The average Mw of the HA component of Thealoz duo^®^ was around 200 kDa [7]. Thus, the formulation contained low-Mw HA. Figure 7A shows that low Mw HA (Mw~300 kDa) in HEPES or trehalose buffer offered the lower protection. While the average Mw of the polymer component of Systane hydration have not decreased after treatment with hyaluronidase, which means that the predominant polymer was hydroxypropyl-guar. In the case of ReNu^®^, the cell survival rate is lower than the positive control indicating the cytotoxicity of the lubricant.

## 4. Conclusions

A friction method was proposed using biologically relevant test conditions (physiological blinking), with the potential to be included in the evaluation routine and characterization of lubricants based on HA. Even though the source of commercial eye drops was characterized by a range of lubricating properties, the coefficient of friction described the lubricating effect. The tribological study demonstrated that medium molecular weight HA (native and modified) was characterized by a lower coefficient of friction than high molecular weight HA. The coefficient of friction might correlate improved clinical performance. The amphiphilic HA2-C12 reduced the friction even at low concentration (0.1% wt). Thus, blurring effect could be dismissed. The chemical modification imparted a strong interaction with mucin III, which could be formulated for contact lenses wearers.

The formulation with trehalose increased interaction between HA and mucin, but it might decrease comfort for contact lenses wearers. HA is efficient in reducing the coefficient of friction (CoF), but the effect is molecular weight-dependent.

## 5. Patents

CZ2019598A3: A method of preparation of an ester derivative of hyaluronic acid having an even substituent distribution, the ester derivative of hyaluronic acid, a composition comprising thereof and use thereof. 

## Figures and Tables

**Figure 1 biomolecules-11-01431-f001:**
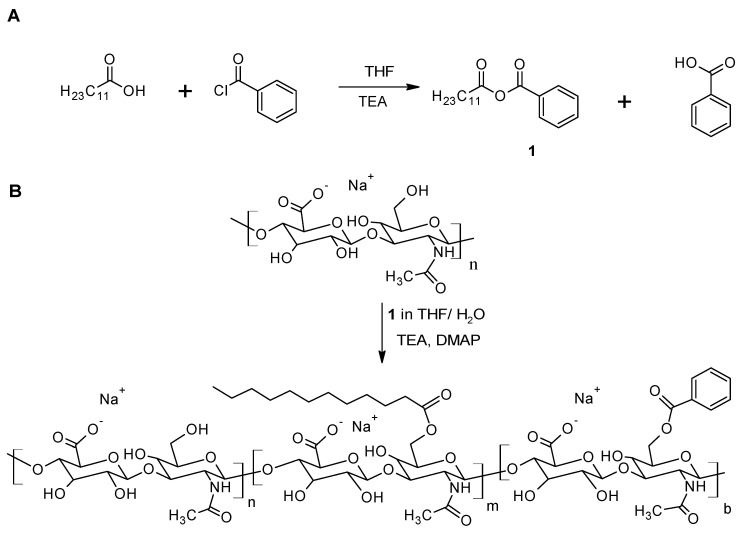
(**A**) The activation of dodecanoic acid mediated by benzoyl chloride. (**B**) The esterification of medium molecular weight HA carried out in water/THF.

**Figure 2 biomolecules-11-01431-f002:**
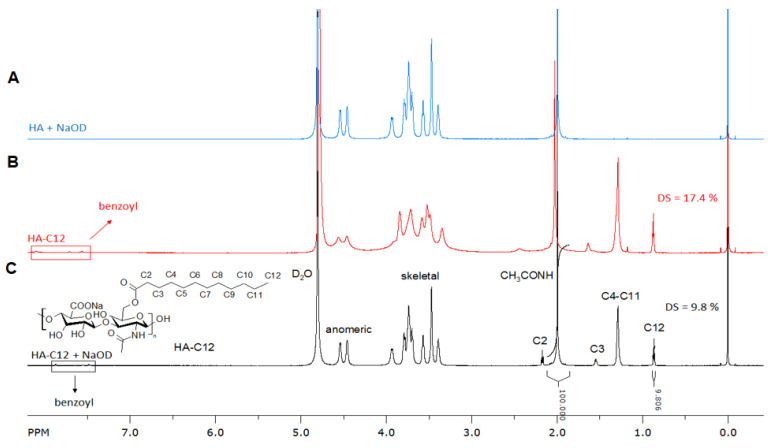
^1^H NMR spectrum of (**A**) Native HA (300 kDa) and esterified HA0-C12 measured in (**B**) D_2_O and (**C**) NaOD.

**Figure 3 biomolecules-11-01431-f003:**
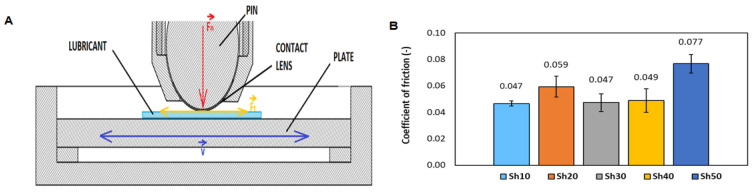
(**A**) Pin-on-plate configuration used in this work (**B**) The average friction coefficient of different silicone ball-hardness at normal force 1 N.

**Figure 4 biomolecules-11-01431-f004:**
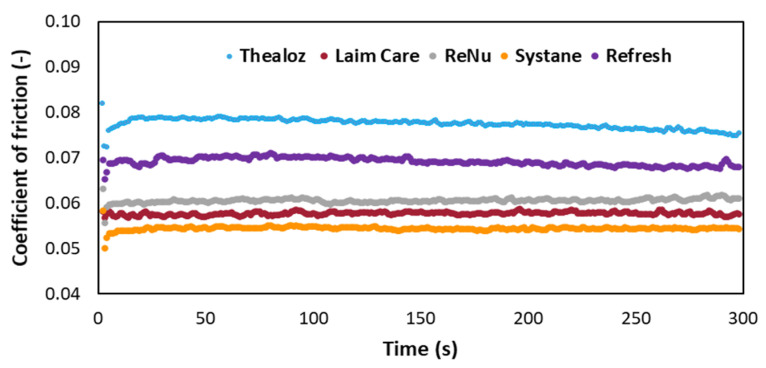
Friction coefficient dependence on time for five commercial eye drops.

**Figure 5 biomolecules-11-01431-f005:**
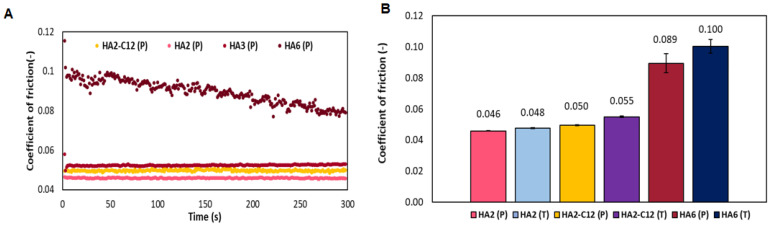
(**A**) Development of friction coefficient for HA and HA2-C12 as a function of time (**B**) Average coefficient of friction of HA and HA2-C12 samples in saline buffer (P). (**B**) Comparison of coefficient of friction for native HA and HA2-C12 in saline (P) or trehalose (T) buffers, respectively.

**Figure 6 biomolecules-11-01431-f006:**
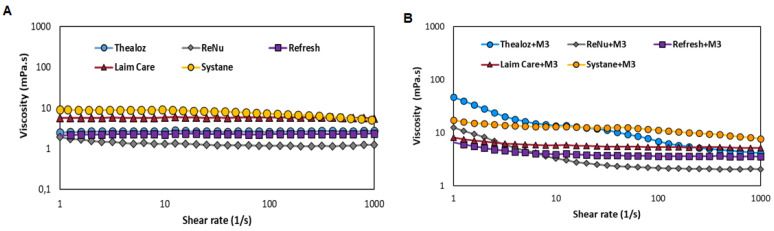
(**A**) Viscosity of eye drops determined as a function of shear rates (**B**) Mucoadhesive properties for eye drops with mucin III evaluated at shear rates 1–1000 s^−1^.

**Figure 7 biomolecules-11-01431-f007:**
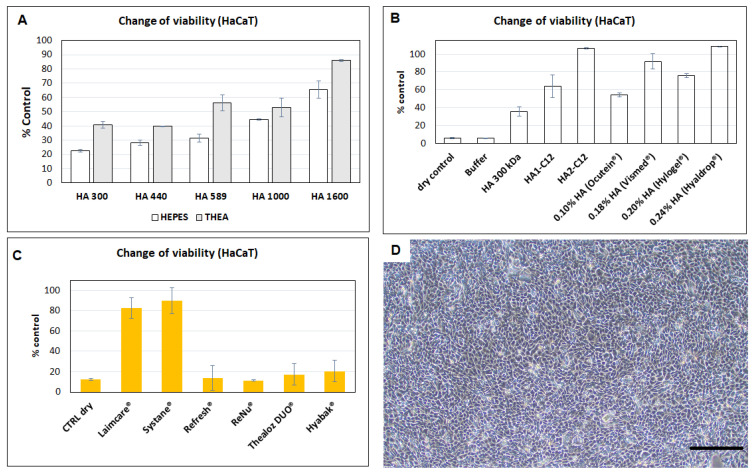
(**A**) The hydration-protection effect against desiccation after pre-treatment for HA (in HEPES or trehalose buffers) 0.3 wt.%. Results are mean ± SD, *n* = 3. (**B**) The hydration-protection effect against desiccation after pre-treatment for HA, HA2-C12 and OTC products. (**C**)The hydration-protection against desiccation after pre-treatment with selected OTC products. The results are an average (mean value ± SD, *n* = 6) Significant differences (*p* < 0.05) were compared to the control “dry cells” (**D**) Optical microscopic imagen of cell protection obtained for HA2-C12 (scale bar = 300 µm).

**Table 1 biomolecules-11-01431-t001:** Sodium dodecanoyl hyaluronan (HA-C12) studied in this work.

Entry ^a^	Sample ^b^	DS_GC_ (%) ^c^	Mw (kDa) ^d^ (PDI)	Dry Matter (*w*/*w*) ^e^ %	ash (*w*/*w*) %	T% ^f^	τ ^g^
1	HA0-C12	9.33 ± 032	199.4 ± 5.9 (1.6)	86.83 ± 0.66	6.6 ± 0.09	99.2 ± 0.21	2.1 ± 0.4
2	HA1-C12	8.73 ± 0.12	239.3 ± 2.9 (1.4)	88.10 ± 0.16	7.4 ± 0.33	98.8 ± 0.30	2.8 ± 0.5
3	HA2-C12	9.03 ± 0.10	311.2 ± 5.2 (1.4)	88.66 ± 1.5	7.8 ± 0.66	98.9 ± 0.44	3.8 ± 0.2
4	HA3-C12	7.73 ± 0.04	434.5 ± 1.9 (1.5)	90.03 ± 1.3	4.9 ± 0.12	94.7 ± 1.4	5.8 ± 1.6

^a^ All the reactions were optimized (1 eq. of fatty acid, 3 eq. of TEA, 0.05 eq. of DMAP). ^b^ Identification of the sample used in this work. ^c^ The degree of substitution determined by gas chromatography. ^d^ The Mw of HA-C12 and the polydispersity index (PDI = Mw/Mn) in brackets. ^e^ The dry matter and ash were determined by thermogravimetric analyses (TGA). ^f^ the transmittance at 660 nm at c = 1% (*w*/*v*). ^g^ τ is the turbidity.

**Table 2 biomolecules-11-01431-t002:** Viscosity values and mucoadhesive index for HA (0.3%) and HA-C12 (0.1–0.3%) determined in the presence of mucin III. The values are reported as an average (mean value ± SD, *n* = 3) at shear rate of 33.9 s^−1^.

Entry	[%]	Sample	Medium	Viscosity (mPa·s)	Viscosity (mPa·s) HA + Mucin III	Mucoadhesive Index (%)
1	-	Mucin III	HEPES	4.97	--	--
2	0.3	HA2	HEPES	14.68 ± 0.01	25.12 ± 3.18	27.83
3	0.3	HA5	HEPES	17.88 ± 0.002	10.07 ± 0.003	−49.8
4	0.3	HA6	HEPES	119.93 ± 0.001	21.30 ± 0.001	−72.6
5	0.3	HA2	trehalose	8.15 ± 0.001	12.73 ± 0.002	−2.95
6	0.3	HA5	trehalose	17.64 ± 0.001	28.48 ± 0.004	25.78
7	0.3	HA6	trehalose	49.33 ± 0.003	69.34 ± 0.004	26.69
8	0.10	HA2-C12	HEPES	3.8 ± 0.04	7.69 ± 0.11	−12.31
9	0.20		HEPES	7.38 ± 0.02	21.65 ± 0.11	75.30
10	0.30		HEPES	12.91 ± 0.19	331.4 ± 37.09	1753.5
11	0.30	HA2-C12	trehalose	16.77 ± 0.003	494.63 ± 0.010	2175.2
12	0.10	HA3-C12	HEPES	6.57 ± 0.75	11.67 ± 0.12	1.12
13	0.20		HEPES	13.53 ± 0.37	58.07 ± 1.06	213.89
14	0.30		HEPES	26.53 ± 0.96	156.83 ± 3.94	397.87
15	0.30	HA3-C12	trehalose	68.87 ± 0.001	226.30 ± 0.002	254.14

## Data Availability

Not applicable.

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
