# Peer review of "Insight into the Lubrication and Adhesion Properties of Hyaluronan for Ocular Drug Delivery"

_biomolecules, 2021, doi:10.3390/biom11101431_

Round 1
Reviewer 1 Report
This manuscript is extremely difficult to read. Many sentences are unclear. Many details are left out, and concepts jump from one to another without apparent connection. It should be thoroughly revised, to improve the English but also to improve logical flow of thought and data-driven analysis and discussion.
The paper should also be cut by half. Keep only the analysis of pure HA of differing molecular weight and concentration in a physiological buffer, in comparison with the commercial eye drop formulations.
The results section for that paper would start with current section 3.4, Tribological model of the eye. Then, measurement of the coefficient of friction for each HA sample could be directly compared with the commercial formulations. Based on the data in Figures 6 and 7, solutions of 440kDa HA and 589 kDa HA at 0.3% concentration have coefficient of friction properties that are similar to Systane and Laim Care, which contain HA. The very high coefficient of friction reported for the (one) higher molecular weight 1656 kDa HA shows that the experimental apparatus is measuring fluid film hydrodynamic lubrication rather than boundary lubrication, and is therefore expected to increase friction with increasing solution viscosity. The statement in the abstract (and the title of the paper) suggest that medium molecular weight HA improves lubrication, and implies it is better than high molecular weight HA, but that would not be true in a boundary lubrication model. The one single sample of high molecular weight HA, analyzed by one method, is not sufficient to support such a broad statement. The article title should be changed.
The paper could then report the "mucoadhesive" properties of HA solutions and commercial eye drops. Unfortunately, Table 3 is difficult to decipher. Does HA2 in HEPES buffer have increased viscosity in the presence of mucin, but HA5 and HA6 have reduced viscosity? What does this mean?
The cell survival data in Figure 9 look interesting, but the description of the method and its validity are too brief to allow firm conclusions.
Overall, the paper comparing pure HA to commercial eye drops would be interesting to a general audience, if more carefully organized and described.
In contrast, the data on HA-C12 is so poorly presented that it carries no interest at this time. The SEC-MALLS are poorly established, and direct comparison of the pure HA with the HA-C12 in the same solvent, with a dn/dc measured in that solvent, would be needed to establish any difference in molecular weight. The transmittance data and turbidity measurements are hardly described, and incorrect footnotes for them are given in Table 1. The NMR and mass spectrometry data need much more detailed analysis. In general, the samples are poorly characterized, in part due to their physical properties.
Reviewer 2 Report
Titel:
- The title needs some improvement from the language point of view.
Abstract:
- The abstract needs to be rewritten. The beginning of the abstract should be shortened and streamlined.
- The use of different times is completely mixed (present tense, past tense, future).
- Overall, the entire abstract reads mixed up and must be improved.
Keywords:
- The choice of the keywords could be more specific.
Introduction:
- HA is important for many biotribological purposes, such as joint lubrication. This should be reflected in the overall introduction (second paragraph). You may refer to a recently published article by Marian et al. “Exploring the Lubrication Mechanisms of Synovial Fluids for Joint Longevity–A Perspective”
- The novelty of the present study should be better worked out.
- The end of the abstract reads confusing. Authors introduce the conducted work before switching again to some literature review. Please streamline.
Experimental:
- How close is the following approximation to reality “A pin (eyeball model) was simulated using 194 a silicone rubber ball (Elastosil® LR 3003, hardness = 10 Sh)”
- How were the friction forces measured in the equipment?
- On which base have you selected a force of 1 N and a frequency of 6 Hz? No real justification has been given.
Results:
- Section 3.1 presents some experimental details, which should be moved to the experimental section.
- The values presented in Table 1 should be presented with mean values and error bars.
- Figure 3 appears to be blurred. Please improve the overall quality of this figure.
- In the part “results and discussion”, more methods are used than described in the experimental section. Please extend.
- The overall readability of Figure 4 is poor. Please improve the quality and readability. How representative are the results?
- Section 3.3 also includes some repetition of experimental details previously mentioned. Please remove.
- How many tribological experimental experiments were conducted to obtain Figure
- Besides the averaged data, it would be recommendable to present the evolution of the COF over time. This would enable more insights into the overall tribological performance of the different materials used.
- What was the acting lubrication regime?
- The manuscript makes use of a lot of sub-headings. To a certain extent, the readability and overall flow of the manuscript suffer from that.
- The conclusions section should be shortened.
Round 2
Reviewer 2 Report
I am fine with the revised version.
Thank you very much for addressing my comments and concerns.